# Contrast Enhanced Mammography (CEM) Enhancing Asymmetry: Single-Center First Case Analysis

**DOI:** 10.3390/diagnostics13061011

**Published:** 2023-03-07

**Authors:** Giuliano Migliaro, Giulia Bicchierai, Pietro Valente, Federica Di Naro, Diego De Benedetto, Francesco Amato, Cecilia Boeri, Ermanno Vanzi, Vittorio Miele, Jacopo Nori

**Affiliations:** 1Breast Imaging Diagnostic Unit, Azienda Ospedaliero-Universitaria Careggi, 50134 Florence, Italy; 2Breast Imaging Diagnostic Unit, Radiology Department, Ospedale San Giovanni di Dio, 92100 Agrigento, Italy; 3Department of Radiology, Azienda Ospedaliero-Universitaria Careggi, 50134 Florence, Italy

**Keywords:** enhancing asymmetry, CEM, breast cancer, second look, breast imaging

## Abstract

(1) Purpose: The latest Breast Imaging Reporting and Data System (BI-RADS) lexicon for CEM introduced a new descriptor, enhancing asymmetries (EAs). The purpose of this study was to determine which types of lesions were correlated with EAs. (2) Methods: A total of 3359 CEM exams, executed at AOUC Careggi in Florence, Italy between 2019 and 2021 were retrospectively assessed by two radiologists. For each of the EAs found, the size, the enhancing conspicuity (degree of enhancement relative to background described as low, moderate, or high), whether there was a corresponding finding in the traditional radiology images (US or mammography), the biopsy results when performed including any follow-up exams, and the presence of background parenchymal enhancement (BPE) of the normal breast tissue (minimal, mild, moderate, marked) were described. (3) Results: A total of 64 women were included, 36 of them underwent CEM for a preoperative staging assessment, and 28 for a problem-solving examination. Among the 64 EAs, 19/64 (29.69%) resulted in being category B5 (B5) lesions, 5/64 (7.81%) as category B3 (B3) lesions, and 40/64(62.50%) were negative or benign either after biopsy or second-look exams or follow-up. We assessed that EAs with higher enhancing conspicuity correlated significantly with a higher risk of B5 lesions (*p*: 0.0071), especially bigger ones (*p*: 0.0274). Conclusions: EAs can relate both with benign and tumoral lesions, and they need to be assessed as the other CEM descriptors, with re-evaluation of low-energy images and second-look exams, particularly larger EAs with higher enhancing conspicuity.

## 1. Introduction

Contrast-enhanced mammography (CEM) uses intravenous iodinated contrast administration to highlight tumoral neovascularization. After the intravenous contrast medium injection, low-energy (LE) and high-energy (HE) images are acquired, and then the post-processing combination of them generates the recombined images (RC) [1].

CEM is advisable for pre-operative assessment of breast cancer, as a work-up examination of suspicious findings, to monitor the response after neoadjuvant chemotherapy and for breast cancer screening in patients with intermediate risk of breast cancer [2,3,4].

In 2022, the American College of Radiology (ACR) published the BI-RADS lexicon for CEM, with the aim of standardizing imaging reporting. Prior to this, radiologists used to apply MRI lexicon to CEM findings [5], while the new BI-RADS lexicon proper to CEM establishes the descriptors to correctly interpret each finding, both in low and recombined images [6].

The descriptors provided by the MRI lexicon are indeed “focus” (a small dot of enhancement not representing a space-occupying lesion), “mass” (a space-occupying lesion with recognizable margins and shape), and “non-mass enhancement” (an area of enhancement that does not meet the criteria for a mass) [7,8]. Conversely, the new BI-RADS lexicon for CEM institutes the terms “mass” (3D space occupying lesion with a convex-outward contour) and “non-mass enhancement” (as for MRI, an area of enhancement that does not meet the criteria for a mass), but it also introduces the “enhancing asymmetries”(EAs), defined as a finding visualized on only one view on the recombined images, either craniocaudal (CC) or mediolateral oblique (MLO) [6].

As far as we know, this is the first study in the literature that assesses the impact on the clinical practice of EA.

The aim of this study, indeed, was to define the incidence of this new finding, which kind of lesions, whether benign or tumoral, EAs are more often correlated with, in order to better shape the management of patients in which these findings are observed.

## 2. Materials and Methods

This is a single-center retrospective study, and the approval by the Institutional review board (IRB) was waived.

A total of 3359 CEM exams was conducted in our Breast Imaging Diagnostic Unit between January 2019 and December 2021.

Our CEMs casuistry was retrospectively reviewed by two radiologists, in consensus, with 5 and 10 years of experience in breast radiology, in search of enhancing asymmetries, defined as a finding seen in only one view on the RC images, either craniocaudal (CC) or mediolateral oblique (MLO), as stated by the brand new ACR BI-RADS lexicon for CEM [6]. There were six radiologists who initially reported the exams, all with various levels of experience in breast radiology, from 5 to 30 years. They also performed the second-look ultrasound (SL-ultra- sound) and second-look DBT (SL-DBT) after CEM, the breast magnetic resonance imaging (MRI) and the biopsies after CEM.

The 64 CEMs from 64 patients were collected where an EA was found. A total of 36/64 (56.25%) underwent CEM for presurgical staging due to a biopsy-proven malignant lesion, 28/64 (43.75%) as the work-up examination of suspicious findings detected with ultrasound (US), digital mammography (DM) or digital breast tomosynthesis (DBT).

The inclusion criteria were CEM exam executed at our center, at least one EA visible in the RC images, radiological second look examinations, and follow-up after CEM performed at our center with reports available on the RIS-PACS system, biopsies, and vacuum assisted excision (VAE) or surgery performed after CEM in our hospital with the availability of post-biopsy or post-surgical reports in the Intranet system of our hospital.

The exclusion criteria were patients who did not perform follow-up or second look after CEM in our department; patients who have had biopsies or post-CEM surgery or VAE in other medical centers.

Hence, the 64 CEM exams of 64 women aged between 34 and 82 years old (mean age 56.7 years old, median age 52 years old) were included in our study.

For all the 64 EAs identified, as suggested by the BI-RADS lexicon, it was evaluated whether there was a correlation on the LE images (as mass asymmetry, focal asymmetry, architectural distortion, or calcifications), and therefore, all patients were subjected to second-look ultrasound (SL-ultrasound) and second-look DBT (SL-DBT) after CEM [9] (Figure 1).

The EAs with a correlate detected on the LE images or by SL-ultrasound and/or SL-DBT and catalogued as either BI-RADS 3, BI-RADS 4, or BI-RADS 5 [10,11] underwent US-guided core needle biopsy (CNB) or tomosynthesis-guided vacuum assisted breast biopsy (VABB).

On the other hand, when a correlate was detected either on the LE images or by SL-ultrasound and/or SL-DBT was classified as BI-RADS 1 or BI-RADS 2 [10,11], patients were subjected to annual follow-up with mammography and US.

The EAs without any correlate underwent MRI with contrast medium due to the current unavailability at our center of CEM-guided biopsy, which would have been preferred [12,13]. The MRIs were classified and managed according to the BI-RADS criteria of breast MRI [7]. Those classified as BI-RADS 1 and 2 were postponed to annual follow-up with MRI; those classified as BI-RADS 3 had a six-month follow-up with MRI, and those classified as BI-RADS 4 or 5 underwent an MRI-guided biopsy (Figure 1).

Lesions resulting in being malignant (BI-RADS 6) after CNB, VABB, or MRI-guided VABB were forwarded to a surgeon for surgical excision; those found to be benign had an instrumental follow-up with mammography and ultrasound at our department, first at six months, then every 12 months; lesions resulting in being B3 were subjected to surgical excision or removal by VAE.

The histologic results obtained from the surgically excised lesions, VAE, or biopsy were considered the gold standard for comparison with the CEM findings in the case of EA with a correlate detected on LE images or by SL-ultrasound and/or SL-DBT. The results of MRI, MRI-guided biopsy, and follow-up were considered the gold standard for comparison with the CEM findings in the case of EA without a correlate detectable on the LE images or by SL-ultrasound and/or SL-DBT.

Furthermore, as suggested by the BI-RADS lexicon and conducted for the other CEM findings such as mass and non-mass enhancement (NME), for every EA found, we depicted the size, the internal enhancing pattern, either homogeneous (a confluent uniform enhancement) or heterogeneous (a not-uniform enhancement in a random pattern), the conspicuity of this enhancement (degree of enhancement relative to background described as low, moderate, or high), the level of background parenchymal enhancement (BPE) of normal breast tissue (minimal, mild, moderate, marked), and the breast composition.

All CEM exams included were performed with the same protocol of execution on the same commercial mammography system (Selenia Dimensions, Hologic, Marlborough, MA, USA). Image acquisition started 2 min after 1.5 cc/kg of body weight iodine-based contrast agent intravenous injection (Iopromide 370 mg mL^−1^; Bayer HealthCare, Whippany, NJ or Iopamidol 370 mg mL^−1^; Bracco Imaging S.p.A., Milan, Italy) was administered at 3 cc/s followed by 20 mL of a saline flush using an automated power injector.

Dual-energy LE and HE of each breast were captured within 5 min in both CC and MLO in sequence to obtain early RC images; additional images, defined as late RC, were acquired 8 min after contrast injection with the same setting to allow for qualitative assessment of the enhancement kinetics.

LE and HE were performed at 26–31 kVp with rhodium and silver (Rh and Ag) filters and at 45–49 kVp with a copper filter, respectively. A recombination algorithm was used to subtract the unenhanced breast tissue and to provide RC [1].

DM and DBT were performed using a full-field digital mammography unit with tomosynthesis (Selenia Dimensions, Hologic, Bedford, MA, USA).

Second-look US exams were conducted using a 10–13 MHz transducer and a US unit (ESAOTE, MyLab 70 XVG, Genoa, Italy). All the MRI examinations were performed in the prone position with dedicated breast coils; 1.5-Tesla equipment was used (Symphony, Siemens Medical System, Erlangen, Germany; Philips Medical System, DA Best, The Netherlands; Magnetom Avanto, Siemens Medical System, Erlangen, Germany).

Stereotactic-guided VAB was performed using a vacuum-assisted biopsy device (Mammotome revolve; Devicor Medical Products) with an 8-gauge needle, and a mean of 8 core samples per lesion was obtained (range 6–10). VABs were made on a digital prone table (Affirm Prone Biopsy System; Hologic). Percutaneous CNB was performed with a semiautomated biopsy gun (Precisa, Hospital Service) with a 14-G, 10-cm-long needle. A mean of three core samples per lesion (range 3–7) was obtained. An MRI-guided biopsy was performed using a coaxial 9-gauge Suros ATEC^®^Breast Biopsy and Excision System (Hologic) with a lateral approach. A mean of 24 samples per lesion was obtained. The samples were analyzed by two pathologists with more than 25 years of experience in breast pathology [14]. The radiological findings and the pathological reports of each patient were then discussed at the multidisciplinary breast meeting group of our hospital.

### Statistical Analysis

The statistical method used to analyze our data was Fisher’s exact test (FET). After all of the biopsies and follow-up exams that allowed us to catalogue our EAs according to the BI-RADS classification, we compared the data related to benign lesions after biopsy or follow-up exams with the data of malignant lesions through the histologic results obtained from the surgically excised lesions, VAE, or biopsy. In detail, we confronted some of the internal characteristics of the EA found such as the type of correlate detected on LE images or by SL-ultrasound and/or SL-DBT, enhancing conspicuity and the enhancing pattern, the size, the degree of BPE and breast composition, aiming to understand if one of these was related to a higher risk of a malignant lesion.

A value of *p* < 0.05 was considered a statistical difference.

All analyses were performed with IBM SPSS Statistics 23.0 (IBM SPSS Inc., Chicago, IL, USA), Microsoft Excel (Microsoft Corporation) © 2020 MedCalc Software Ltd. (Ostend, Belgium).

## 3. Results

After reviewing the casuistry of CEMs executed at our center, we retrospectively collected 64 EAs of 64 patients (1.91% of the 3359 CEMs). The population consists of 64 women aged between 34 and 82 years old (mean age 56.75 years old, median age 52 years old), 36/64 (56.25%) underwent CEM for presurgical staging due to biopsy-proven lesions, and 28/64 (43.75%) as work-up examination after previous inconclusive exams.

Our statistical analysis showed that a low enhancing conspicuity was related to a lower risk for an EA to be malignant (*p*: 0.0117), and, in the same way, a high conspicuity was related to a higher risk of a B5 biopsy-proven lesion (*p*: 0.0071), especially the larger the EA (0.0274) (Table 1) (Figure 2a,b).

Of the 64 EAs found, 45 underwent biopsy (70.31%), either CNB, VABB, or MRI-guided, and the results were respectively: 18/64 (28.13%) B5 lesions, 5/64 (7.81%) B3 lesions, and 22/64 (34.38%) B2 lesions; the remaining 19/64 (28.23%) were not biopsied: 2/19 underwent MRI whose findings were classified as B2 lesions [7]; 16/19 corresponded to a not-suspicious correlate detected on LE images or found by SL-ultrasound and/or SL-DBT and were classified as BI-RADS 1 or BI-RADS 2, then subjected to annual follow-up; 1/19 underwent CEM as pre-surgical staging due to a BI-RADS 5 lesion, already eligible to mastectomy, was directly subjected to surgery, and the surgical specimens confirmed a multifocal malignant lesion, allowing us to define our EA as an additional malignant lesion.

In total, the overall results were as follows: 19/64 (29.7%) malignant lesions and 45/64 (70.3%) benign lesions.

In detail, we detected 19 B5 lesions, among them, four were biopsied before CEM and underwent the exam as presurgical staging of these biopsy-proven malignant index lesions; four were biopsied after CEM, where they were found and initially subjected as a problem-solving examination, and, only together with SL-ultrasound and/or SL-DBT, were we allowed to correctly spot these hidden lesions. The remaining 10 lesions were biopsied after CEM because they had been detected in CEMs executed as pre-surgical staging of other biopsy-proven malignant index lesions. After SL-ultrasound and/or SL-DBT, we were able to detect biopsy and thus retrieve these additional lesions. One lesion was not biopsied because the patient who underwent CEM as a pre-surgical staging of a biopsy-proven malignant index lesion was already eligible to mastectomy, and the surgical specimens confirmed a multifocal malignant lesion, allowing us to define our EA as an additional malignant lesion.

Amid the malignant lesions, 10/19 resulted in G1 cancers, in particular, four invasive tubular carcinomas, three invasive ductal carcinomas, one ductal and papillary carcinoma, one lobular carcinoma, one colloid carcinoma; 5/19 lesions were G2 cancers, one ductal and papillary carcinoma, one invasive ductal carcinoma, two ductal in situ carcinomas, one invasive lobular carcinoma; 4/19 resulted as G3 cancers, two invasive ductal carcinomas, one lobular and ductal carcinoma, and one ductal and colloid carcinoma (Table 2).

With the intrinsic molecular subtype, the malignant lesions were distinguished as 10 luminal A (52.6%), eight luminal B (42.2%), one basal-like (5.2%), and no Her-2 like tumor was detected (Table 3).

Among the benign lesions detected after biopsy (27/45), we found five B3 lesions (two atypical ductal hyperplasias, one radial scar, one classical lobular neoplasia, one flat epithelial atypia), and 22 B2 lesions, the most frequently found was fibrocystic breast change (16/22) (Table 2).

As above-mentioned, for each EA found, we described the size, the internal enhancing pattern, the conspicuity of this enhancement, the patient’s breast composition, and the level of BPE [6].

In detail, the overall mean size was 9.56 mm [3–40 mm], the mean size of the biopsy-proven malignant lesions was 12.05 mm [4–40 mm] (median size 8 mm); the mean size of the not-B5 lesions (weather B3, B2, lesions without a correlate) was 8.51 mm [3–23 mm] (median size 8 mm).

We hence detected how the malignant EA’s maximum sizes were significantly greater than the sizes of EAs not related to the B5 lesions (*p* = 0.0274; 95 CI% 0.4081–6.6749). We subdivided the population according to the maximum sizes in two groups (<10 mm and >10 mm): using the Fisher exact test (FET), no statistically significant difference emerged between the two groups (Table 4).

The internal enhancing pattern was reported as homogeneous for 35/64 (54.68%) and as heterogeneous in the remaining 29 EAs (45%). From the perspective of benignity and malignity of the lesions found, in 13/19 malignancies, the enhancement was more intense and heterogeneous, in contrast to 6/19 of them where the internal pattern was homogeneous; as far as the benignities found, 29 had a homogeneous internal enhancing pattern while 16 had a heterogeneous one [16,17] (Table 4).

We also described the patient breast composition: 4/64 were BI-RADS A (almost entirely fatty), 36/64 were classified as BI-RADS B (scattered areas of fibroglandular density), 15/64 as BI-RADS C (heterogeneously dense), and 9/64 as BI-RADS D (extremely dense) [10].

With regard to the conspicuity of the enhancement, that is, the degree of enhancement relative to the background, it was outlined for each EA either as low, moderate, or high.

Specifically, 26/64 EAs were interpreted to have a low conspicuity, 20/64 a moderate conspicuity, and 18/64 a high one (Table 4).

Among the B5 lesions, 3/19 resulted in having a low conspicuity, 6/19 moderate conspicuity and 10/19 a high conspicuity whereas the conspicuity of the not-B5 lesions is reported as follows: 23/45 as low, 14/45 as moderate, and 8/45 as high, and as already mentioned previously, the statistical analysis showed us that a low enhancing conspicuity is related to a lower risk for an EA to be malignant (*p*: 0.0117) and, in the same way, high conspicuity is related to a higher risk of a B5 biopsy-proven lesion (*p*: 0.0071) (Table 4) (Figure 3 and Figure 4).

The resulting level of enhancing conspicuity using the Fisher exact test (FET) was related to various histological types, intrinsic molecular subtype, or tumor grading level, but any statistically significant difference was detected (Table 4).

Since it can be an obstacle to spot the EAs and increased BPE has increased odds for breast cancer [18], the BPE was also described, which was distinguished into minimal, mild, and moderate and marked accordingly, as already stated, to the level of background parenchymal enhancement of the normal tissue of each breast where the EAs had been found.

Among the 64 EAs collected, BPE was minimal in 23/64 patients, mild in 20/64, moderate in 13/64, and marked in 8/64 (Table 4).

Regarding the possible correlates of the EA, we categorized the findings in major categories: focal asymmetrical densities (6/64); architectural distortions (8/64); microcalcifications (12/64); opacity (19/64) seen on DM and/or DBT; masses with indistinct margins (6/64) and hypoechoic masses with circumscribed margins (10/64) detected with US [11]; masses, non-masses or foci (3/64) shown on MRI [7] (Table 4, Figure 5 and Figure 6).

Except for the statistically significant differences found and already mentioned, all the remaining EA characteristics listed above (such as internal enhancing pattern, breast composition and BPE) did not show any significantly higher risk of correlating with a malignant lesion.

Within all suspicious correlates (46/64) found either on the LE images, detected by SL-ultrasound and/or SL-DBT or MRI when no correlate was found on conventional imaging and analyzed via the histologic results obtained from the surgically excised lesions, VAE, or biopsy. Eventually, we retrieved 22/64 BI-RADS 2 lesions, 5/64 BI-RADS 3 lesions, and 19/64 BI-RADS 5 (Table 2).

On the other hand, among all the not-suspicious correlates that had not been biopsied, we collected 18/64 between the BI-RADS 1 and BI-RADS 2 lesions (Table 2).

## 4. Discussion

A retrospective study reviewed 3359 CEM exams accomplished at our Breast Imaging Diagnostic Unit between January 2019 and December 2021 in search of EAs, defined as a finding seen on only one view on the RC images, either CC or MLO, as stated by the brand new ACR BI-RADS lexicon for CEM [6]. We collected 64/3359 (1.9%) CEM of 64 different patients where an EA was found. To our knowledge, so far, this is the first work to analyze this new CEM category introduced with the BI-RADS lexicon; in particular, in addition to the incidence of EA, we tried to understand if there were any characteristics more frequently associated with EAs corresponding to lesions resulting in histologically malignant and others associated with EAs resulting in benign lesions as well as to understand what is the best procedure to manage these findings.

The incidence of EAs in our series was not very high (1.9%), as, to date, there are no other works published on the subject. It would be more appropriate to wait for studies on an even larger scale to have these data confirmed.

Of the 64 EAs found, 45 underwent biopsy (70.31%), either CNB, VABB, or MRI-guided, 19/64 (28.23%) were not biopsied because 16/19 corresponded to a correlate classified as BI-RADS 1 or BI-RADS 2, 2/19 underwent MRI whose findings were classified as BI-RADS 2 lesions [7], 1/19, who underwent CEM as pre-surgical staging due to a BI-RADS 5 lesion, already eligible for mastectomy who was directly subjected to surgery and the surgical specimens confirmed a multifocal malignant lesion, allowing us to our define our EA as an additional malignant lesion.

Among the EAs without a correlate, three underwent MRI and one of these even MRI-guided biopsies, since CEM-guided biopsy was not available at that time at our center. However,, it would have been preferred to better target the EA to put them under biopsy. As a matter of fact, there have been preliminary studies supporting CEM-guided biopsy compared to an MRI-guided one, as in R. Alcantara et al., who first stated it, and Schiaffino S. and Cozzi A., who later emphasized that CEM-guided breast biopsy had success and complication rates similar to those reported for magnetic resonance guidance, ensuring fast, low-cost, and effective tissue sampling of the CEM-detected lesions [12,13].

Regarding how many correlate-less EAs were recorded (3/64, 4.7%), this study matched what has already been described by Viggiano T et al., who stated that enhancing findings seen only on subtraction CEM images were seen in 4% of cases in clinical practice [19].

Eventually, 45 (70.31%) EAs resulted in being benign after histological examination or follow-up and 19 (29.69%) were malignant after biopsy or post-surgical histological examination. Among the 19 malignant lesions, the most frequently detected was ductal invasive carcinoma (9/19) luminal B (with Ki67 >20%) [20].

Altogether, focusing on EAs detected through CEM exams executed as pre-surgical staging, we were allowed to find 11 additional malignant lesions corresponding to 11 EAs, undetectable on conventional imaging, in addition, among the CEMs motivated as problem-solving exams, we were able to detect four malignant lesions not visible on traditional imaging: as a result of this, we could depict a total of 15 malignant lesions that may otherwise have been overlooked.

These data are in line with what has already been published by Bicchierai G. et al., regarding the percentage of the impact of CEM in discovering additional lesions [14].

Even considering how preliminary our results are with regard to detecting an EA, due to our outcome in terms of the number of malignant lesions retrieved, we strongly recommend to keep investigating this topic with work-up exams or biopsy, and not classify it as a non-specific finding.

This study shows that an EA with low enhancement intensity is more frequently related to benign alterations than one with high intensity, which appears to be associated with malignant lesions. This result is in line with what has already been published in the literature first by Luczynska et al. or by Thomas Knogler et al. and then by Chi X et al. [16,21,22], whose work showed that medium or strong enhancement on CEM was the most likely indicator of malignancy while benign lesions most frequently had a medium or weak enhancement. However, in both our work and in those above-mentioned, the intensity of enhancement was qualitatively evaluated, even though studies with quantitative measurements of intensity in CEM have been published such as those of Ying Liu et al. and Rudnicki W et al., whose results, recently supported by Savaridas SL et al., actually confirm the ones of the qualitative studies (i.e., that infiltrating cancers showed the highest values, while benign lesions the lowest) [17,23,24].

This study had some limitations due to the small number of patients included, the retrospective setting, and the main issue related to the qualitative and not quantitative assessment of enhancing conspicuity.

## 5. Conclusions

Although this is a preliminary experience and the data will be confirmed on a larger scale, it is reasonable to state that when an EA is detected, especially if not small and with high conspicuity, it is strongly recommended to subject it to the same diagnostic path followed for other descriptors such as masses and non-masses, which means to re-evaluate low-energy images and keep investigating it with work-up second-look exams and biopsies or, in the case of an EA without a correlate, with MRI and MRI biopsy, or even better, with CEM biopsy.

## Figures and Tables

**Figure 1 diagnostics-13-01011-f001:**
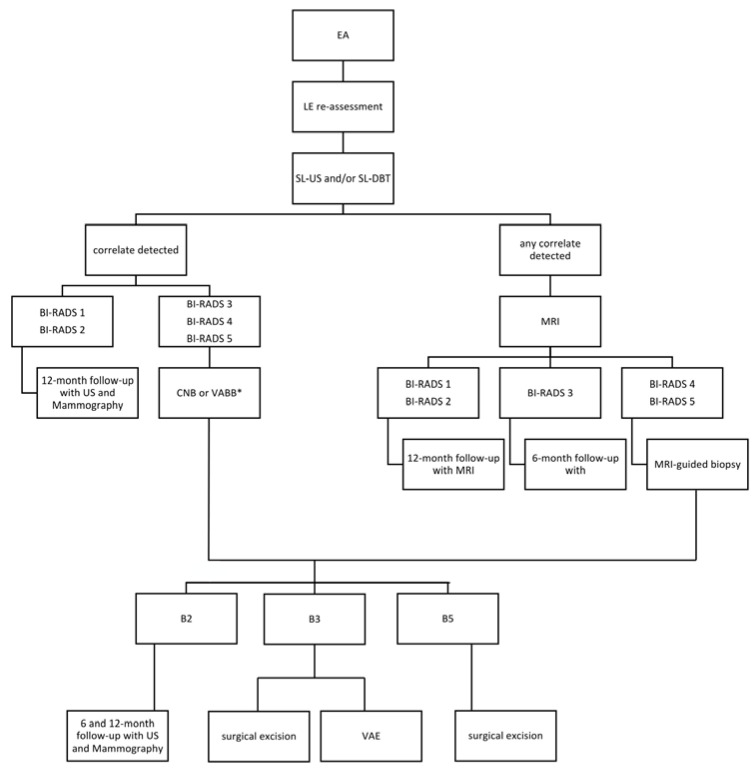
Flowchart of the diagnostic management of CEM enhancing asymmetries—LE: low-energy images; SL: second-look; US: ultrasound; DBT: digital breast tomosynthesis; BI-RADS, Breast Imaging Reporting and Data Systems; CNB: US-guided core needle biopsy (CNB); VABB: tomosynthesis-guided vacuum assisted breast biopsy; MRI: magnetic resonance imaging; VAE: vacuum assisted excision; * Depending on how they have been detected.

**Figure 2 diagnostics-13-01011-f002:**
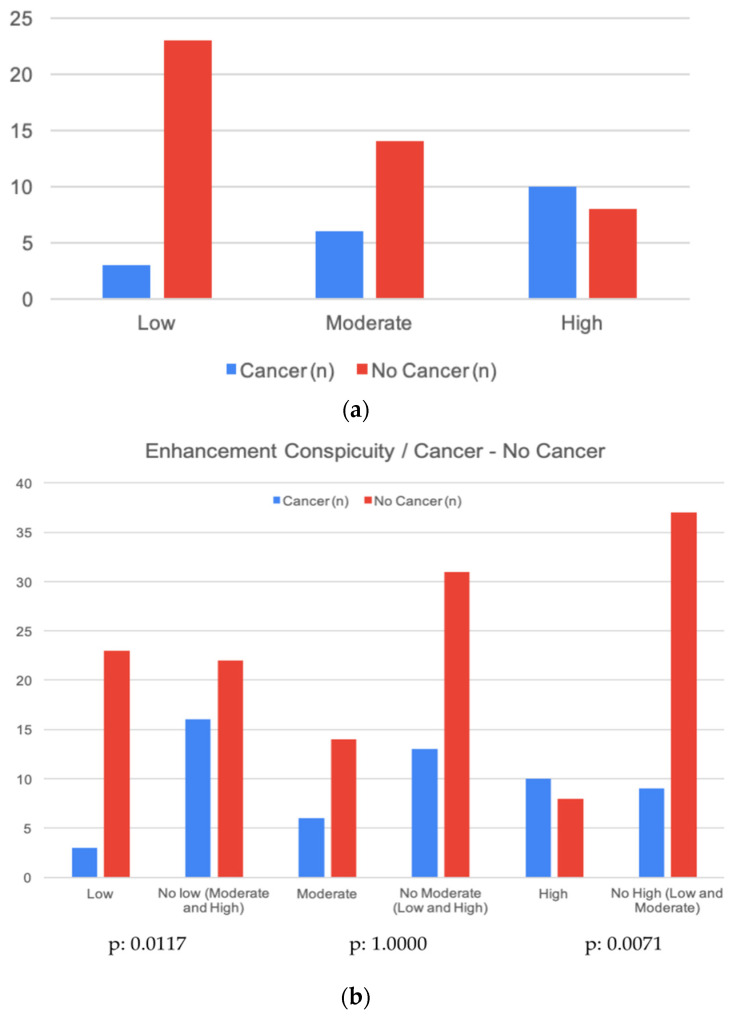
(**a**,**b**) Graphic representation of the contingency table related to the enhancement conspicuity and the risk of cancer [15].

**Figure 3 diagnostics-13-01011-f003:**
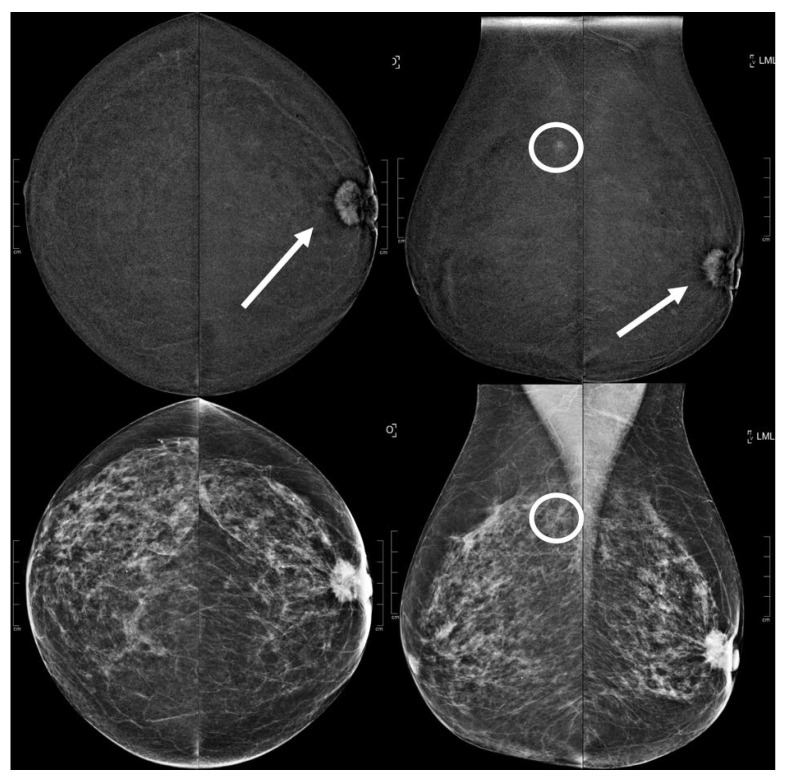
A 73 year old patient. Contrast-enhanced digital mammography performed as preoperative staging for a B5 lesion in the left breast (arrows). In the early RC in the right breast, there was an enhancing asymmetry in the upper quadrants with high conspicuity only visible in MLO (circle). The EA did not have a clear correspondence in low-energy images. The patient underwent SL-US, which showed a hypoechoic mass of 6.0 mm that was subsequently subjected to CNB (core needle biopsy), which confirmed an invasive tubular carcinoma.

**Figure 4 diagnostics-13-01011-f004:**
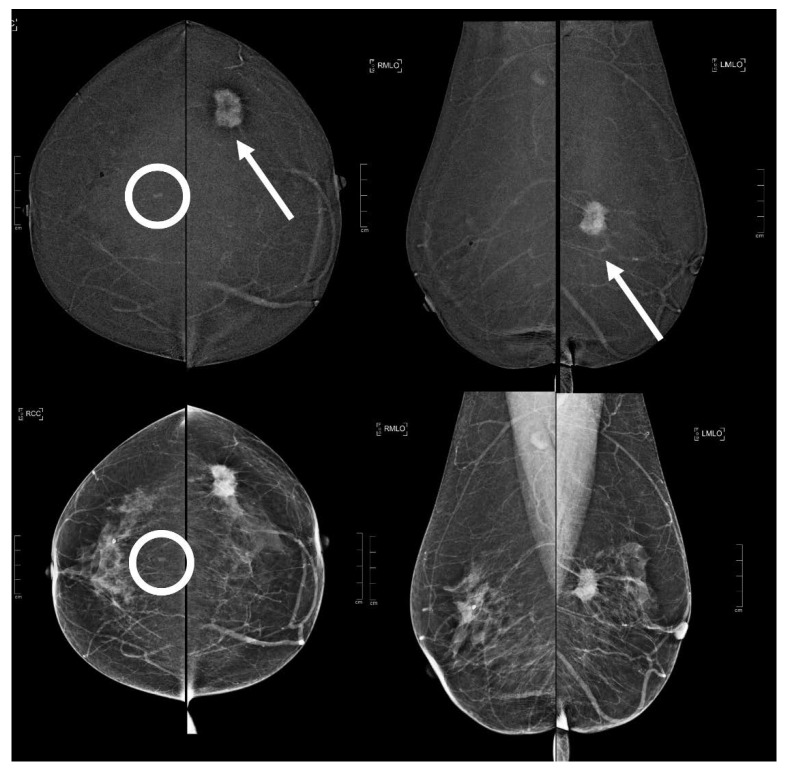
A 62 year old patient. Contrast-enhanced digital mammography performed as preoperative staging for a B5 lesion in the left breast (arrows). In the early RC in the right breast, there was an enhancing asymmetry in the central quadrants with moderate conspicuity only visible in CC (circle), corresponding to an opacity in low-energy images. This opacity subsequently underwent VABB (tomosynthesis-guided vacuum assisted breast biopsy), which showed a benign lesion (fibrocystic breast change).

**Figure 5 diagnostics-13-01011-f005:**
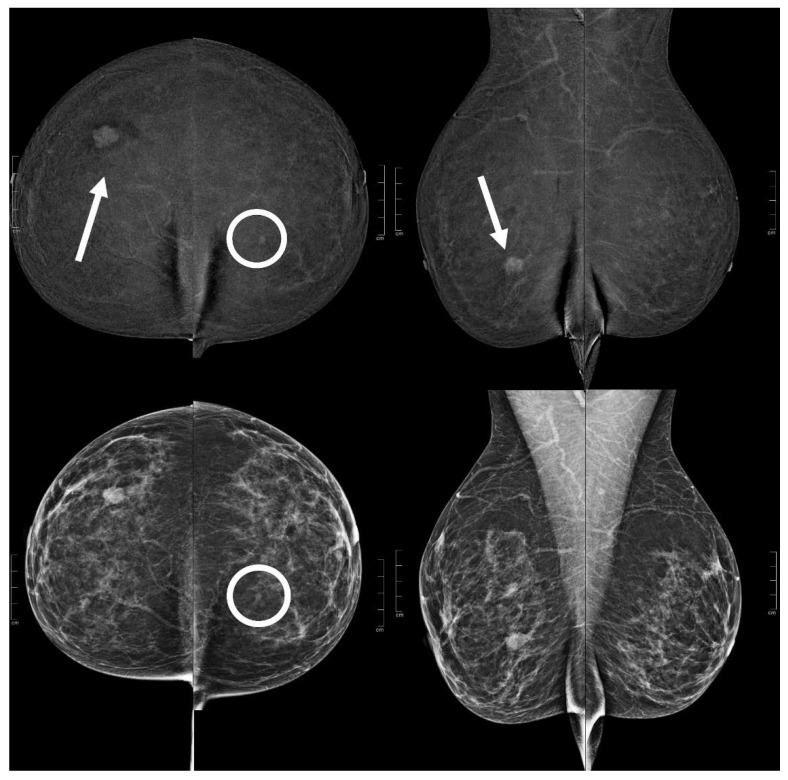
A 51 year old patient. Contrast-enhanced digital mammography performed as preoperative staging for a B5 lesion in the right breast (arrows). In the early RC in the left breast, there was an enhancing asymmetry in the inner quadrants with moderate conspicuity only visible in CC (circle), corresponding to an opacity in low-energy images. This opacity subsequently underwent VABB (tomosynthesis-guided vacuum assisted breast biopsy), which confirmed an invasive carcinoma.

**Figure 6 diagnostics-13-01011-f006:**
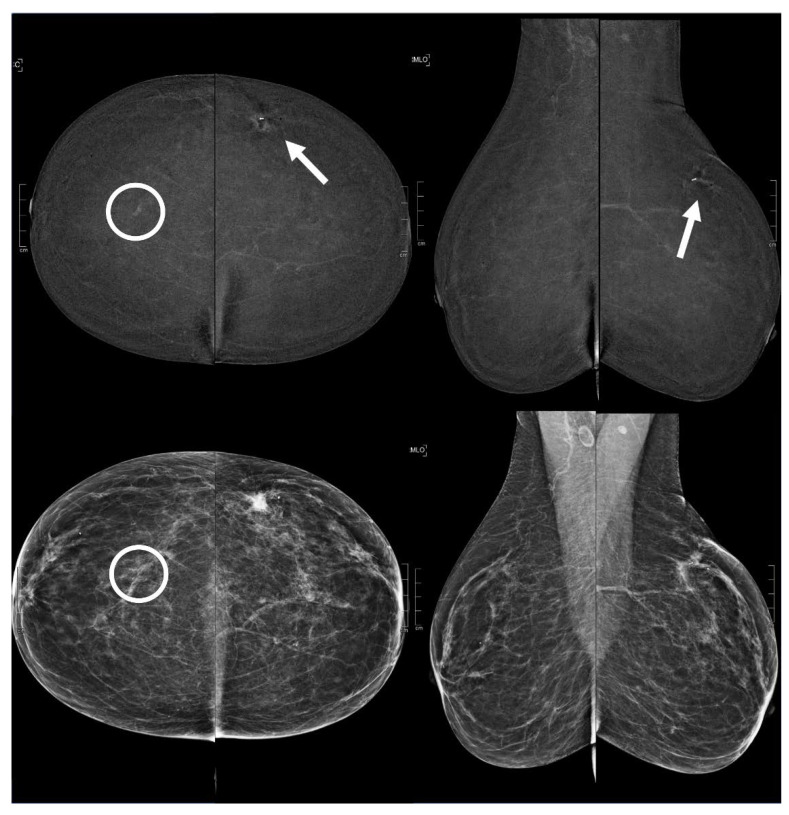
A 60 year old patient. Contrast-enhanced digital mammography performed as preoperative staging for a B5 lesion in the left breast (arrows). In the early RC in the right breast, there was an enhancing asymmetry in the central quadrants with high conspicuity only visible in CC (circle), corresponding to an architectural distortion in low-energy images. This architectural distortion subsequently underwent VABB (tomosynthesis-guided vacuum assisted breast biopsy), which confirmed a ductal in situ carcinoma.

**Table 1 diagnostics-13-01011-t001:** The contingency table related to the enhancement conspicuity and the risk of cancer [15].

Fisher Exact Test—Enhancement Conspicuity/Cancer-No Cancer			
**Enhancement conspicuity**	**Cancer (n)**	**No Cancer (n)**	***p*-value**
Low	3	23	*0.071*
Moderate	6	14	
High	10	8	
**Enhancement conspicuity**	**Cancer (n)**	**No Cancer (n)**	***p*-value**
Low	3	23	*0.0117*
Not Low (Moderate and High)	16	22	
**Enhancement conspicuity**	**Cancer (n)**	**No Cancer (n)**	***p*-value**
Moderate	6	14	*1.0000*
Not Moderate (Low and High)	13	31	
**Enhancement conspicuity**	**Cancer (n)**	**No Cancer (n)**	***p*-value**
High	10	8	*0.0071*
Not High (Low and Moderate)	9	37	

**Table 2 diagnostics-13-01011-t002:** List of 64 EAs detected with each histopathological results, catalogued according to the core biopsy reporting categories, with the prevalence of the type of lesion found; the B5 lesions were also subdivided based on the histological grading of the malignancies detected.

B5			
	GRADING	HYSTOLOGY	
	**G1 (10/19)**		
		Invasive tubular carcinoma	4
		Invasive ductal carcinoma	3
		Ductal and papillary carcinoma	1
		Lobular carcinoma	1
		Colloid carcinoma	1
	**G2 (5/19)**		
		Ductal and papillary carcinoma	1
		Invasive ductal carcinoma	1
		Ductal in situ carcinoma	2
		Invasive lobular carcinoma	1
	**G3 (4/19)**		
		Invasive ductal carcinoma	2
		Lobular and ductal carcinoma	1
		Ductal and colloid carcinoma	1
* **TOTAL** *			* **19** *
**B3**		Atypical ductal hyperplasia	2
		Radial scar	1
		Classical lobular neoplasia	1
		Flat epithelial atypia	1
* **TOTAL** *			* **5** *
**B2**		Fibrocystic breast change	16
		Fibroadenoma	2
		Phlogosis	2
		Sclerosing adenosis	1
		Bening adenomyoepithelioma	1
* **TOTAL** *			* **22** *

**Table 3 diagnostics-13-01011-t003:** Summary of the 19 malignant EAs included in the study, detected after SL-US and/or SL-DBT or MRI and subjected to biopsy, each with the patient’s age, characteristics of the onset (if principal or additional lesion), final histopathological results, and other specifics for each lesion (grading, HER2 status, molecular subtype).

Patient Age	Additional/Principal	Size (mm)	Histology Results after Surgery	Grading	Her2 Status	Molecular Subtype
82	Additional	7	Colloid carcinoma	G1	Neg.	LUM B
48	Additional	6	Invasive ductal carcinoma	G1	Neg.	LUM A
70	Principal	4	Invasive ductal carcinoma	G1	Neg.	LUM A
59	Additional	9	Lobular carcinoma	G1	Neg.	LUM A
73	Additional	3	Invasive tubular carcinoma	G1	Neg.	LUM A
77	Additional	4	Invasive tubular carcinoma	G1	Neg.	LUM B
78	Additional	5	Invasive tubular carcinoma	G1	Neg.	LUM A
73	Additional	7	Invasive ductal carcinoma	G1	Neg.	LUM B
73	Principal	4	Ductal and papillary carcinoma	G1	Neg.	LUM A
53	Principal	11	Invasive tubular carcinoma	G1	Neg.	LUM A
46	Principal	4	Ductal in situ carcinoma	G2	Neg.	LUM A
60	Additional	5	Ductal in situ carcinoma	G2	Neg.	LUM A
53	Principal	11	Invasive ductal carcinoma	G2	Neg.	LUM B
46	Principal	9	Ductal and papillary carcinoma	G2	Neg.	LUM B
71	Principal	14	Invasive lobular carcinoma	G2	Neg.	LUM A
51	Additional	3	Invasive ductal carcinoma	G3	Neg.	Basal-like
42	Additional	15	Ductal and colloid carcinoma	G3	Neg.	LUM B
48	Principal	2.5	Invasive ductal carcinoma	G3	Neg.	LUM B
62	Additional	9	Lobular and ductal carcinoma	G3	Neg.	LUM B

**Table 4 diagnostics-13-01011-t004:** Characteristics of the patients and radiological features of the EAs included in the study, divided based on benignity and malignancy; benignity was proven via biopsy or assessed according to the BI-RADS level of suspicion after SL-US and/or SL-DBT, MRI, or follow-up exams; malignancy was proven via biopsy.

	Patients	EA		*p* Value
		Benign (%)	Malignant (%)	
**INDICATION**				
Pre-surgical assessment	36	23 (63.9%)	13 (36.1%)	*ns*
Problem-solving	28	22 (78.6%)	6 (21.4%)	*ns*
**BREAST COMPOSITION**				
A	4	1 (25%)	3 (75%)	*ns*
B	36	24 (66.7%)	12 (33.3%)	*ns*
C	15	13 (86.7%)	2 (13.3%)	*ns*
D	9	7 (77.8%)	2 (22.2%)	*ns*
**BPE**				
Minimal	23	16 (69.6%)	7 (30.4%)	*ns*
Mild	20	12 (60%)	8 (40%)	*ns*
Moderate	13	10 (76.9%)	3 (23.1%)	*ns*
High	8	7 (87.5%)	1 (12.5%)	*ns*
**CORRELATE**				
Focal Density	6	4 (66.7%)	2 (33.3%)	*ns*
Opacity	19	14 (73.7%)	5 (26.3%)	*ns*
Microcalcifications	12	11 (91.7%)	1 (8.3%)	*ns*
Architectural distortion	8	3 (37.5%)	5 (62.5%)	*ns*
US hypoechoic masses with circumscribed margins	10	5 (50%)	5 (50%)	*ns*
US masses with indistinct margins	6	5 (83.3%)	1 (16.7%)	*ns*
MRI mass/non-mass/focus	3	3 (100%)	0	*ns*
**BIOPSY**				
Percutaneous CNB	12	6 (50%)	6 (50%)	
Stereotactic VABB	32	20 (62.5%)	12 (37.5%)	
MRI-biopsy	1	1 (100%)	0	
No biopsy	19	18 (94.7%)	1 (5.3%)	
**PRINCIPAL/ADDITIONAL**				
Principal	19	11 (57.9%)	8 (42.1%)	*ns*
Additional	45	34 (75.6%)	11 (24.4%)	*ns*
**SIZE**				
Mean (mm)		8.51 [3–23]	12.05 [4–40]	*p: 0.0274*
Median (mm)		8	8	*ns*
**INTERNAL ENHANCING PATTERN**				
Homogeneous	35	29 (82.9%)	6 (17.1%)	*ns*
Heterogeneous	29	16 (55.2%)	13 (44.8%)	*ns*
**ENHANCING CONSPICUITY**				
Low	26	23 (88.5%)	3 (11.5%)	*p: 0.0117*
Moderate	20	14 (70%)	6 (30%)	*ns*
High	18	8 (44.4%)	10 (55.6%)	*p: 0.0071*

## Data Availability

Not applicable.

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
