# Peer review of "Contrast Enhanced Mammography (CEM) Enhancing Asymmetry: Single-Center First Case Analysis"

_diagnostics, 2023, doi:10.3390/diagnostics13061011_

Round 1

Reviewer 1 Report

I'm afraid I have to disagree this is the first study of the EA applicate to CEM. Such as the paper last Dec, “Diagnostic value of contrast-enhanced mammography in the characterization of breast asymmetry.”

Some digit in the paper uses a comma instead of a dot; please correct them.

Please provide one or two examples of the correlations plot with its p-value.

Any single view asymmetries cases in your study?

Author Response

Response to Reviewer 1 Comments

Point 1: I'm afraid I have to disagree this is the first study of the EA applicate to CEM. Such as the paper last Dec, “Diagnostic value of contrast-enhanced mammography in the characterization of breast asymmetry.”

Response 1: Thank you for your kind and accurate suggestion. We carefully read the study you proposed, we agree that we both studied asymmetries but we actually noted some differences.

For istance, the main one is about what we consider as asymmetry and what they do. We define “asymmetry” as a finding visualized on only one view on the recombined images of the contrast-enhanced mammography (CEM), either craniocaudal or mediolateral oblique, as it was recently defined by the latetes BI-RADS lexicon for CEM (2022). On the other hand, they consider focal, developing and global asymmetries, which are findings visible on two projections. Moreover, they refer to Breast Imaging-Reporting and Data System (BI-RADS) of mammography to define the “asymmetry” they included. Hence, even though they included also single view asymmetries, those are mammographic asymmetries, and the authors did not precise if they correspond to an enhancing asymmetry in CEM. While - as already mentioned - our “asymmetries” are the ones seen in CEMs.

Furthermore, they state several times that the aim of their study was to assess the diagnostic role of CEM in the characterization of breast asymmetries detected on mammograms. Indeed, their inclusion criteria considered patients having (before CEM) indeterminate and suspicious asymmetries on mammography with and without an ultrasound correlate, which later underwent CEM to precisely study them.

Conversly our starting point is different, as we refer to the latest BI-RADS lexicon for CEM (2022), which introduces the enhancing asymmetry as a new descriptor for CEM and our purpose is to define the incidence of this new finding and which kind of lesions the enhancing asymmetry more often correlates with, re-evaluating low-energy images and subjecting patients to other exams (breast ultrasound, thomosyntehsis, MRI, biopsy). Conclusively, our inclusion criteria are women undergoing CEM as a presurgical staging of biopsy proven malignant lesions or as a work-up examinations and the asymmetry we study are the enchancing asymmetry that emerge in those CEMs, not asymmetrical findings found before the exam on mammography.

For all the reasons above we would appreciate to keep affirming that this is the first study in literature that assesses the impact on the clinical practice of the new descriptor “enhancing asymmetry”.

Point 2: Some digit in the paper uses a comma instead of a dot; please correct them.

Response 2: Thank you for your comment. We modified the text as you suggested, here an example from the Results “Of 64 EAs found, 45 underwent biopsy (70.31%), either CNB, VABB, or MRI-guided and the results were respectively: 18/64 (28.13%) B5 lesions, 5/64 (7.81%) B3 lesions and 22/64 (34.38%) B2 lesions; the remaining 19/64 (28.23%) weren’t biopsied: 2/19 underwent MRI whose findings were classified as B2 lesions”

Point 3: Please provide one or two examples of the correlations plot with its p-value.

Response 3: Thank you for your comment. Here is the example you asked for, we report the contingency tables related to the enhancement conspicuity and the risk of cancer.

Fisher Exact Test – Enhancement Conspicuity / Cancer-No Cancer

Enhancement conspicuity

Cancer (n)

No Cancer (n)

p-value

Low

3

23

0,071

Moderate

6

14

High

10

8

Enhancement conspicuity

Cancer (n)

No Cancer (n)

p-value

Low

3

23

0,0117

No Low (Moderate and High)

16

22

Enhancement conspicuity

Cancer (n)

No Cancer (n)

p-value

Moderate

6

14

1,0000

No Moderate (Low and High)

13

31

Enhancement conspicuity

Cancer (n)

No Cancer (n)

p-value

High

10

8

0,0071

No High (Low and Moderate)

9

37

Reference

Chong Sun Hong, Tae Gyu Oha Correlation plot for a contingency table Communications for Statistical Applications and Methods  2021, Vol. 28, No. 3, 295–305

Point 4: Any single view asymmetries cases in your study?

Response 4: Thank you for your comment. Among the EAs included, 45 have a mammographic correlate, unfortunately all of them are visible on two projections (unlike the corresponding EAs, which are visible either on CC projection or on MLO projection)

Reviewer 2 Report

Abstract: line 10: please expand BI-RADs term to Breast Imaging Reporting and Data System at first appearance in the text.

Line 11:  'This study aim to.....' better to be expresses as The purpose of this study is to determine which types of lesions are associated/correlated  with EAs

line 17 Expand BPE term also.

Line 19-20  'as B5 lesions'  please mentioned this as category B5. Just mentioned the word Category.

Introduction: Full stop or comma should be after citation (reference No please check this point for most citation here at introduction section and most other section.

Line 40-43 which represents the study importance , please edit this phrase well and add suitable reference citation to this part.

Line 47: aim of the study, attempt to change the word relate to correlate.

Materials and Methods 

Line 52: 'A retrospective study.......'  this is repeated meaning see line 50 please start with the number of examinations directly.

Results

Line 174 'showed us that a low' remove pronounce. 

The headline of the table should appear above the table. and headline of the figure should appear under the figure.

Tables and figures are of acceptable quality

Discussion is good

Author Response

Response to Reviewer 2 Comments

Point 1: Abstract: line 10: please expand BI-RADs term to Breast Imaging Reporting and Data System at first appearance in the text.

Response 1: Thank you for your suggestion, we changed the text as follows: “Purpose: The latest Breast Imaging Reporting and Data System (BI-RADS) lexicon for CEM introduced a new descriptor, the Enhancing Asymmetry (EAs)”

Point 2: Line 11:  'This study aim to.....' better to be expresses as The purpose of this study is to determine which types of lesions are associated/correlated  with EAs

Response 2: Thank you, we changed it as you suggested: “The purpose of this study is to determine which types of lesions are correlated with EAs”

Point 3: line 17 Expand BPE term also.

Response 3: Thank you for the comment, we modified it as you suggested: “…the presence of background parenchymal enhancement (BPE) of normal breast tissue…”

Point 4: Line 19-20  'as B5 lesions'  please mentioned this as category B5. Just mentioned the word Category. 

Response 4: Thank you for your suggestion, we modified the text as follow : resulted as category B5 (B5) lesions, 5/64 (7,81%) as category B3 (B3) lesions

Point 5: Introduction: Full stop or comma should be after citation (reference No please check this point for most citation here at introduction section and most other section.

Response 5: Thank you for your comment, we corrected the citations as you suggested both in the introduction and in the other section. Example line 33 “…generates the recombined images (RC) [1].”

Point 6: Line 40-43 which represents the study importance , please edit this phrase well and add suitable reference citation to this part.

Response 6: Thank you for your comment, we made the following changes: “In 2022, the American College of Radiology (ACR) published the BI-RADS lexicon for CEM, with the aim of standardizing imaging reporting. Prior to that, radiologists used to apply MRI lexicon to CEM findings [5], while the new BI-RADS lexicon proper to CEM establishes the descriptors to correctly interpret each finding both in low and recombined images [6].

The descriptors provided by the MRI lexicon are indeed “focus” (a small dot of enhancement not representing a space-occupying lesion), “mass” (a space-occupying lesion with recognizable margins and shape) and “non-mass enhancement” (an area of enhancement that does not meet the criteria for a mass) [7,8]. Conversely the new BI-RADS lexicon for CEM institutes the terms “mass” (3D space occupying lesion with a convex-outward contour) and “non-mass enhancement” (as for MRI, an area of enhancement that does not meet the criteria for a mass ) but it also introduces the “Enhancing Asymmetry”(EAs), defined as a finding visualized on only one view on the recombined images, either craniocaudal (CC) or mediolateral oblique (MLO) [6].

As far as we know, this is the first study in literature that assesses the impact on the clinical practice of EA.

Referencesces:

  1. Kamal RM, Helal MH, Mansour SM, Haggag MA, Nada OM, Farahat IG, Alieldin NH. Can we apply the MRI BI-RADS lexicon morphology descriptors on contrast-enhanced spectral mammography? Br J Radiol. 2016 Aug;89(1064):20160157. doi: 10.1259/bjr.20160157. Epub 2016 Jun 21. PMID: 27327403; PMCID: PMC5124889.
  2. Lee CH, Phillips J, Sung JS, Lewin JM, Newell MS. ACR BI-RADS® ATLAS-MAMMOGRAPHY CONTRAST ENHANCED MAMMOGRAPHY (CEM) (A Supplement to ACR BI-RADS® Mammography 2013) 2022.
  3. Morris EA, Comstock CE, Lee CH, et al. ACR BI-RADS® Magnetic Resonance Imaging. In: ACR BI-RADS® Atlas, Breast Imaging Reporting and Data System. Reston, VA, American College of Radiology; 2013. 
  4. Spak DA, Plaxco JS, Santiago L, Dryden MJ, Dogan BE. BI-RADS® fifth edition: A summary of changes. Diagn Interv Imaging. 2017 Mar;98(3):179-190. doi: 10.1016/j.diii.2017.01.001. Epub 2017 Jan 25. PMID: 28131457.

Point 7: Line 47: aim of the study, attempt to change the word relate to correlate.

Response 7: Thank you for your comment, we changed as you suggested “…EA is more often correlated with…”

Point 8: Line 52: 'A retrospective study.......'  this is repeated meaning see line 50 please start with the number of examinations directly.

Response 8: Thank you, we started as you indicated “3359 CEM exams was conducted…”

Point 9: Line 174 'showed us that a low' remove pronounce. 

Response 9: Thank you, we removed it “Our statistical analysis showed that…”

Point 10: The headline of the table should appear above the table. and headline of the figure should appear under the figure.

Response 10: Thank you we changed the positions of the headlines and we put them above each table.